# Analysis of SARS-CoV-2 RNA Persistence across Indoor Surface Materials Reveals Best Practices for Environmental Monitoring Programs

Rodolfo A. Salido,[a] Victor J. Cantú,[a] Alex E. Clark,[b] Sandra L. Leibel,[c,d] Anahid Foroughishafiei,[a] Anushka Saha,[a] Abbas Hakim,[e] Alhakam Nouri,[e] Alma L. Lastrella,[e] Anelizze Castro-Martínez,[e] Ashley Plascencia,[e] Bhavika K. Kapadia,[e] Bing Xia,[e] Christopher A. Ruiz,[e] Clarisse A. Marotz,[c,e] Daniel Maunder,[e] Elijah S. Lawrence,[e] Elizabeth W. Smoot,[e] Emily Eisner,[e] Evelyn S. Crescini,[e] Laura Kohn,[f] Lizbeth Franco Vargas,[e] Marisol Chacón,[e] Maryann Betty,[c,e,g] Michal Machnicki,[e] Min Yi Wu,[e] Nathan A. Baer,[e] Pedro Belda-Ferre,[c,e] Peter De Hoff,[d,e,h] Phoebe Seaver,[e] R. Tyler Ostrander,[e] Rebecca Tsai,[c,e] Shashank Sathe,[d,e,i] Stefan Aigner,[d,e,i] Sydney C. Morgan,[d,h] Toan T. Ngo,[e] Tom Barber,[e] Willi Cheung,[d,e,j] Aaron F. Carlin,[b] Gene W. Yeo,[d,i] Louise C. Laurent,[d,h] Rebecca Fielding-Miller,[f] Rob Knight[a,c,k,l]

aDepartment of Bioengineering, University of California San Diego, La Jolla, California, USA
bDivision of Infectious Diseases and Global Public Health, Department of Medicine, University of California San Diego School of Medicine, La Jolla, California, USA
cDepartment of Pediatrics, University of California San Diego, La Jolla, California, USA
dSanford Consortium of Regenerative Medicine, University of California San Diego, La Jolla, California, USA
eExpedited COVID Identification Environment (EXCITE) Laboratory, Department of Pediatrics, University of California San Diego, La Jolla, California, USA
fHerbert Wertheim School of Public Health, University of California, La Jolla, California, USA
gRady Children's Hospital, San Diego, California, USA
hDepartment of Obstetrics, Gynecology, and Reproductive Sciences, University of California San Diego, La Jolla, California, USA
iDepartment of Cellular and Molecular Medicine, University of California San Diego, La Jolla, California, USA
jSan Diego State University, San Diego, California, USA
kDepartment of Computer Science and Engineering, University of California San Diego, La Jolla, California, USA
lCenter for Microbiome Innovation, Jacobs School of Engineering, University of California San Diego, La Jolla, California, USA

Rodolfo A. Salido and Victor J. Cantú contributed equally to this work. Author order was determined by relative contribution to high throughput environmental monitoring method development.

**ABSTRACT** Environmental monitoring in public spaces can be used to identify surfaces contaminated by persons with coronavirus disease 2019 (COVID-19) and inform appropriate infection mitigation responses. Research groups have reported detection of severe acute respiratory syndrome coronavirus 2 (SARS-CoV-2) on surfaces days or weeks after the virus has been deposited, making it difficult to estimate when an infected individual may have shed virus onto a SARS-CoV-2-positive surface, which in turn complicates the process of establishing effective quarantine measures. In this study, we determined that reverse transcription-quantitative PCR (RT-qPCR) detection of viral RNA from heat-inactivated particles experiences minimal decay over 7 days of monitoring on eight out of nine surfaces tested. The properties of the studied surfaces result in RT-qPCR signatures that can be segregated into two material categories, rough and smooth, where smooth surfaces have a lower limit of detection. RT-qPCR signal intensity (average quantification cycle [$Cq$]) can be correlated with surface viral load using only one linear regression model per material category. The same experiment was performed with untreated viral particles on one surface from each category, with essentially identical results. The stability of RT-qPCR viral signal demonstrates the need to clean monitored surfaces after sampling to establish temporal resolution. Additionally, these findings can be used to minimize the number of materials and time points tested and allow for the use of heat-inactivated viral particles when optimizing environmental monitoring methods.

**IMPORTANCE** Environmental monitoring is an important tool for public health surveillance, particularly in settings with low rates of diagnostic testing. Time between

**Citation** Salido RA, Cantú VJ, Clark AE, Leibel SL, Foroughishafiei A, Saha A, Hakim A, Nouri A, Lastrella AL, Castro-Martínez A, Plascencia A, Kapadia BK, Xia B, Ruiz CA, Marotz CA, Maunder D, Lawrence ES, Smoot EW, Eisner E, Crescini ES, Kohn L, Franco Vargas L, Chacón M, Betty M, Machnicki M, Wu MY, Baer NA, Belda-Ferre P, De Hoff P, Seaver P, Ostrander RT, Tsai R, Sathe S, Aigner S, Morgan SC, Ngo TT, Barber T, Cheung W, Carlin AF, Yeo GW, Laurent LC, Fielding-Miller R, Knight R. 2021. Analysis of SARS-CoV-2 RNA persistence across indoor surface materials reveals best practices for environmental monitoring programs. mSystems 6:e01136-21. https://doi.org/10.1128/mSystems.01136-21.

Address correspondence to Rebecca Fielding-Miller, rfieldingmiller@health.ucsd.edu, or Rob Knight, robknight@eng.ucsd.edu.

sampling public environments, such as hospitals or schools, and notifying stakeholders of the results should be minimal, allowing decisions to be made toward containing outbreaks of coronavirus disease 2019 (COVID-19). The Safer At School Early Alert program (SASEA) (https://saseasystem.org/), a large-scale environmental monitoring effort in elementary school and child care settings, has processed >13,000 surface samples for SARS-CoV-2, detecting viral signals from 574 samples. However, consecutive detection events necessitated the present study to establish appropriate response practices around persistent viral signals on classroom surfaces. Other research groups and clinical labs developing environmental monitoring methods may need to establish their own correlation between RT-qPCR results and viral load, but this work provides evidence justifying simplified experimental designs, like reduced testing materials and the use of heat-inactivated viral particles.

**KEYWORDS** COVID-19, RT-qPCR, SARS-CoV-2, environmental monitoring, heat-inactivated, surface sampling, swab

Development and characterization of methods for environmental monitoring of severe acute respiratory syndrome coronavirus 2 (SARS-CoV-2) remain important areas of research for identifying and mitigating potential outbreaks as the global pandemic continues. Environmental monitoring offers indirect detection of possibly infectious individuals through noninvasive sampling. In spaces with relatively consistent occupants, detection of SARS-CoV-2 from environmental samples can help identify coronavirus disease 2019 (COVID-19)-infected individuals, ideally before further transmission. Environmental monitoring can also alert public health leadership to the potential presence of an infection even in settings with low diagnostic testing uptake, allowing for the implementation of enhanced nonpharmaceutical interventions (i.e., double masking, increased hand hygiene, improved ventilation efforts) even in the absence of positive diagnostic tests.

SARS-CoV-2 particles are shed by symptomatic and asymptomatic carriers (1) and have been detected on various surfaces (2–5). Viral signatures have been demonstrated to persist up to 4 weeks in bulk floor dust collected from a room with a quarantined individual (6). Previous environmental monitoring studies have detected SARS-CoV-2 on surfaces contaminated by infected individuals in hospitals and congregate care facilities (6–10). Thus, indoor surface sampling can be valuable for detection of infected persons indoors, where transmission risk is highest (11). The Safer At School Early Alert program (SASEA) (https://saseasystem.org/) uses environmental monitoring and collected over 13,000 surface swabs, but we need more information to clarify what these data are telling us over time.

We sought to characterize temporal dynamics underlying detection of SARS-CoV-2 signals from surface swabs from a variety of common indoor surface types using reverse transcription-quantitative PCR (RT-qPCR). The Centers for Disease Control and Prevention (CDC) maintains that the risk of fomite transmission of SARS-CoV-2 is low (12). This study makes no claims of attempting to understand the possibility of or mechanisms behind infection of virus transmitted by fomites but rather on whether and how negative and positive RT-qPCR detection from surface swabs can enable decision-making in outbreak mitigation, focused clinical testing of individuals, and safe reopening of high-traffic, public spaces.

We used RT-qPCR to detect heat-inactivated viral particles on nine surface materials and monitored the persistence of the heat-inactivated virus for 7 days. Each material—acrylic, steel, glass, ceramic tile, melamine-finished particleboard (MFP), painted drywall, vinyl flooring, and two different carpets (olefin and polyester)—was divided into 5-cm by 5-cm grids, and each 25-cm$^2$ square surface of the grid was inoculated with 10 $\mu$l of either a dilution series of heat-inactivated SARS-CoV-2 particles or water. The eight-point dilution series was based on viral genomic equivalents (GEs) as measured by digital droplet PCR (ddPCR). The inoculum dried for 1 h before swabbing. Every 24 h

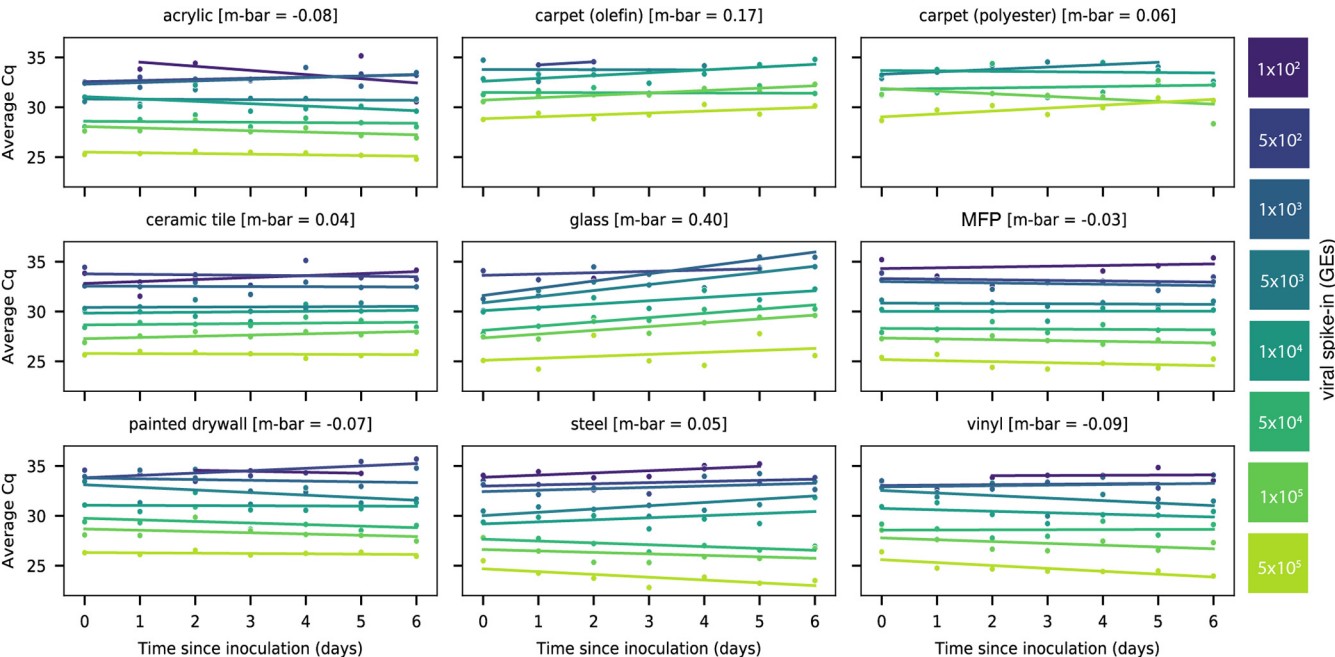

**FIG 1** Scatterplots showing the average *Cq* values of RT-qPCR viral gene calls for corresponding heat-inactivated viral spike-in over 7 days. Viral spike-in concentrations reported as GEs from ddPCR. Linear regressions of average *Cq* values on days since inoculation per spike-in were overlaid on the measured data. The average decay slope (m-bar) is reported alongside each surface type.

postinoculation, an unswabbed section of each material grid was sampled, for a total of 7 days, including the initial postinoculation swab.

To determine whether use of heat-inactivated viral particles in testing and validating environmental monitoring methods reflects results obtained using untreated virus, we compared detection of heat-inactivated SARS-CoV-2 (strain WA-1, SA-WA1/2020) and of authentic, untreated SARS-CoV-2 (variant of concern Beta, isolate B.1.351, hCoV-19/USA/MD-HP01542/2021) on two materials under biosafety level 3 (BSL-3) conditions.

**Findings.** Linear regression of signal intensity (average *Cq* of viral gene calls) on elapsed time since inoculation (days) for each dilution showed minimal decay of viral RNA on eight of nine surface types over 6 days (Fig. 1). The average decay slope for each surface type (m-bar) did not differ significantly from zero (mean = 0.0407, standard deviation [SD] =0.210). RT-qPCR signal decayed with time only on glass (m-bar = 0.401, SD =0.212, differing from the population mean by >1.5 standard deviations).

A two-way repeated measure analysis of variance (ANOVA) on viral signal intensity (average *Cq*) revealed that surface type explains more observed variation in *Cq* than does time since inoculation at the highest concentration ($5 \times 10^5$ GEs) (Fig. 2A). A Kruskal-Wallis *H* test confirmed that mean *Cq*s differ significantly across surface types ($H = 60.86$, $P = 2.49 \times 10^{-9}$) (Fig. 2B), but not across days since inoculation ($H = 1.34$, $P = 0.97$) (Fig. 2C). Pairwise Mann-Whitney *U* tests comparing ranked values of *Cq*s from samples grouped by surface type highlight that both carpet materials (olefin and polyester) are significantly different, after correcting for multiple comparisons (false discovery rate [FDR]-Benjamini/Hochberg, alpha = 0.005), from all other surfaces, but not from each other (Fig. 2B). Other pairwise, significant differences between materials are summarized in Table S1 in the supplemental material. A clustermap of the *U* statistic from the pairwise comparisons effectively clusters samples by material properties, with rough surfaces clustering away from smooth ones (Fig. 2D).

Because RT-qPCR signal intensity for most surfaces was time-invariant, time-collapsed linear regression models relating viral spike-in concentration (log₂ spike-in) to average *Cq* act as standard curves for estimating viral load on different monitored surfaces from *Cq*. After segregating samples based on the qualitative material categories of smooth or rough, linear regressions aggregating all time points yielded one

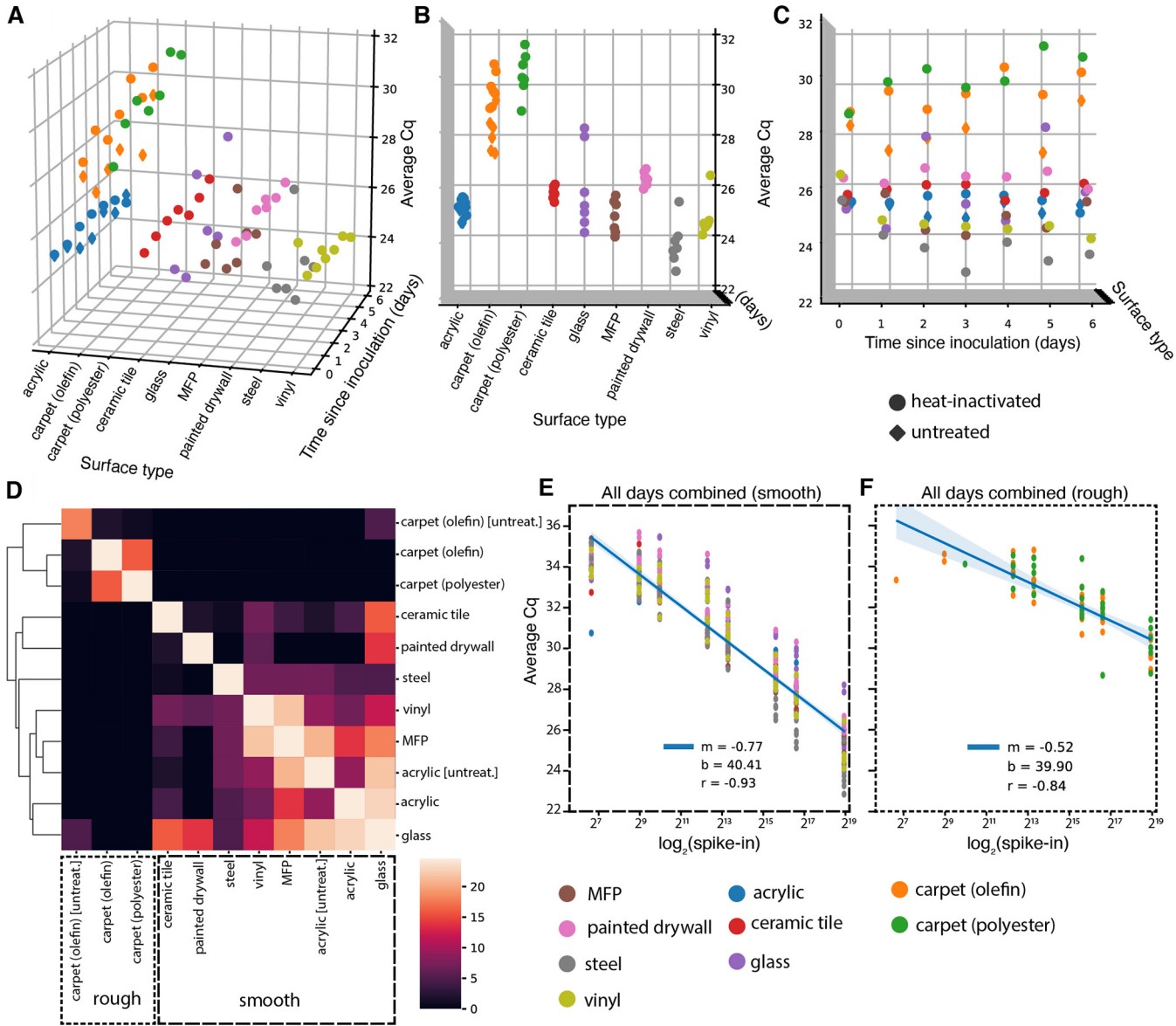

**FIG 2** (A to C) 3D scatterplots showing distribution of average *Cq* values of viral gene calls over 7 days for nine different surfaces inoculated with 5 × 10⁵ GEs (nine surfaces for heat-inactivated virus [circles], two surfaces [acrylic and olefin carpet] for infectious virus [diamonds]). The distribution of *Cq*s differs significantly across surface types (B), but not across days since inoculation (C). (D) Clustermap of the *U* statistic from pairwise Mann-Whitney *U* tests between surface types. (E and F) Standard curves relating surface viral load (log₂ spike-in) to average *Cq* values across all time points for smooth (E) and rough (F) surface types.

standard curve for smooth surfaces ($m = -0.77$, $b = 40.41$, $r = -0.93$) (Fig. 2E) and another for rough surfaces ($m = -0.52$, $b = 39.90$, $r = -0.84$) (Fig. 2F). The reduced slope of the latter curve stems from higher loss of spiked-in viral signal to the rough surface matrix.

To ensure that viral signal stability was not a consequence of selection for resilient viral particles through heat inactivation, we repeated a subset of experiments using infectious virus (untreated) in a BSL-3 laboratory using the B.1.351/Beta variant of SARS-CoV-2 originally identified in South Africa. Due to space limitations in the BSL-3 facility, the untreated virus experiment only included two surface types, acrylic and carpet (olefin) but used the same dilution series and sampling plan.

Results from untreated and heat-inactivated virus are concordant. Untreated virus samples cluster with respect to surface type rather than virion status (heat inactivated or untreated) (Fig. 2D). When evaluating acrylic and carpet (olefin) samples alone, a

mSystems®

Kruskal-Wallis *H* test shows significant differences in the means of *Cq*s across all groups when samples are grouped by surface type ($H = 16.37$, $P = 0.00095$) (see Fig. S1A in the supplemental material), but not when grouped by virion status ($H = 1.96$, $P = 0.161$) (Fig. S1B). Furthermore, linear regression on *Cq* from paired samples between the heat-inactivated and untreated virus experiments show nearly exact correlation despite the use of different variants ($m = 1.05$, $r = 0.97$) (Fig. S1C).

**Discussion.** We show that detecting SARS-CoV-2 RNA on indoor surfaces in environments potentially exposed to COVID-19-infected individuals is effective across a variety of surfaces and a range of initial viral loads. Our swabbing and RT-qPCR methods have greater sensitivity from smooth surfaces (such as MFP—commonly found on desktops—or vinyl flooring) than rough surfaces (carpet). The stability of the viral signal across time limits the ability to estimate when the surface was inoculated but demonstrates that signal can be detected a week postexposure. There is a possibility that viral signal could decay over a longer period of time, but because the motivation behind this study was to improve temporal resolution over shorter periods, this was beyond the scope of the present work. To improve temporal resolution, surfaces swabbed for environmental monitoring should be cleaned with soap and water, following CDC recommendations (13), in order to remove viral signals (12). Previous work with comparable methods for SARS-CoV-2 detection from surfaces demonstrated that washing contaminated objects with household dishwashing detergent for ≥1 min removed enough viral RNA traces so that only 20% of the severely contaminated objects had detectable viral RNA. Furthermore, the average viral load of the washed surfaces was reduced by ~2.5 *Cq*s in comparison to untreated objects (14). Thus, cleaning monitored surfaces with soap and water improves the probability of distinction between persistent or separate exposures in subsequent SARS-CoV-2 detection events.

Although direct inoculation of surfaces with viral particles does not represent interaction with an infected individual in a real-world scenario, we do directly show that untreated and heat-inactivated SARS-CoV-2 particles have similar detectability and stability across surface types. These findings allow the use of heat-inactivated particles in testing and validating environmental monitoring methods and remove the burden of performing such experiments in BSL-3 laboratories.

## SUPPLEMENTAL MATERIAL

Supplemental material is available online only.
**TEXT S1**, DOCX file, 0.02 MB.
**FIG S1**, TIF file, 5 MB.
**FIG S2**, TIF file, 6.6 MB.
**TABLE S1**, DOCX file, 0.01 MB.
**TABLE S2**, DOCX file, 0.01 MB.
**TABLE S3**, DOCX file, 0.01 MB.
**TABLE S4**, DOCX file, 0.01 MB.

## ACKNOWLEDGMENTS

We thank our partner schools and citizen scientists at 15 sites across five districts in San Diego county.

This research was supported by NIH grant (K08AI130381) and a Career Award for Medical Scientists from the Burroughs Wellcome Fund to A.F.C., NIH grant (K01MH112436) to R.F.-M., and the County of San Diego Health and Human Services Agency (contract 563236). This work was performed with the support of the Genomics and Sequencing Core at the UC San Diego Center for AIDS Research (P30 AI036214), the VA San Diego Healthcare System, and the Veterans Medical Research Foundation. The following reagent was deposited by the Centers for Disease Control and Prevention and obtained through BEI Resources, NIAID, NIH: SARS-related coronavirus 2, isolate USA-WA1/2020, NR-52281. The following reagent was obtained through BEI Resources, NIAID, NIH: SARS-related coronavirus 2, Isolate hCoV-19/South Africa/KRISP-K005325/2020, NR-54009, contributed by Alex Sigal and Tulio de Oliveira.

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
