## [Reviewer comments · mSystems]

Analysis of SARS-CoV2 RNA persistence across indoor surface materials reveals best practices for environmental monitoring programs

Rodolfo Salido, Victor Cantú, Alex Clark, Sandra Leibel, Anahid Foroughishafiei, Anushka Saha, Abbas Hakim, Alhakam Nouri, Alma Lastrella, Anelizze Castro-Martínez, Ashley Plascencia, Bhavika Kapadia, Bing Xia, Christopher Ruiz, Clarisse (Lisa) Marotz, Daniel Maunder, Elijah Lawrence, Elizabeth Smoot, Emily Eisner, Evelyn Crescini, Laura Kohn, Lizbeth Franco Vargas, Marisol Chacón, Maryan Betty, Michal Machnicki, Min Yi Wu, Nathan Baer, Pedro Belda-Ferre, Peter De Hoff, Phoebe Seaver, R. Ostrander, Rebecca Tsai, Shashank Sathe, Stefan Aigner, Sydney Morgan, Toan Ngo, Tom Barber, Willi Cheung, Aaron Carlin, Gene Yeo, Louise Laurent, Rebecca Fielding-Miller, and Rob Knight

Corresponding Author(s): Rob Knight, University of California, San Diego

Review Timeline:

Submission Date:

September 21, 2021

Accepted:

October 15, 2021

Editor: Marta Gaglia

Reviewer(s): The reviewers have opted to remain anonymous.

Transaction Report:

DOI: <https://doi.org/10.1128/mSystems.01136-21>

October 15, 2021

Dr. Rob Knight
University of California, San Diego
La Jolla

Re: mSystems01136-21 (Analysis of SARS-CoV2 RNA persistence across indoor surface materials reveals best practices for environmental monitoring programs)

Dear Dr. Rob Knight:

The reviewers agreed that you have thoroughly addressed their comments and that the objective and value of the manuscript is much clearer in the way the phrasing has been changed. Your manuscript has been accepted, and I am forwarding it to the ASM Journals Department for publication. For your reference, ASM Journals' address is given below. Before it can be scheduled for publication, your manuscript will be checked by the mSystems senior production editor, Ellie Ghatineh, to make sure that all elements meet the technical requirements for publication. She will contact you if anything needs to be revised before copyediting and production can begin. Otherwise, you will be notified when your proofs are ready to be viewed.

As an open-access publication, mSystems receives no financial support from paid subscriptions and depends on authors' prompt payment of publication fees as soon as their articles are accepted. =

Publication Fees:

We recognize that the video files can become quite large, and so to avoid quality loss ASM suggests sending the video file via <https://www.wetransfer.com/>. When you have a final version of the video and the still ready to share, please send it to Ellie Ghatineh at eghatineh@asmusa.org.

Sincerely,

Marta Gaglia
Editor, mSystems

Journals Department
Table S3: Accept

Table S1: Accept

Figure S1: Accept

Table S2: Accept

Materials and Methods: Accept

Table S4: Accept

Figure S2: Accept